# Impact of Having a Planned Additional Operation at Time of Liver Transplant on Graft and Patient Outcomes

**DOI:** 10.3390/jcm9020608

**Published:** 2020-02-24

**Authors:** Shirin Salimi, Keval Pandya, Vinay Sastry, Claire West, Susan Virtue, Mark Wells, Michael Crawford, Carlo Pulitano, Geoffrey W. McCaughan, Avik Majumdar, Simone I. Strasser, Ken Liu

**Affiliations:** 1Australian National Liver Transplant Unit, Royal Prince Alfred Hospital, Sydney NSW 2050, Australia; shirin.salimi1@gmail.com (S.S.); kevalvpandya@gmail.com (K.P.); rvinaysastry@gmail.com (V.S.); claire.west1@health.nsw.gov.au (C.W.); susan.virtue@health.nsw.gov.au (S.V.); mark.wells@health.nsw.gov.au (M.W.); michael.crawford1@health.nsw.gov.au (M.C.); carlo.pulitano@gmail.com (C.P.); g.mccaughan@centenary.org.au (G.W.M.); avik.majumdar@health.nsw.gov.au (A.M.); simone.strasser@health.nsw.gov.au (S.I.S.); 2Sydney Medical School, University of Sydney, Sydney NSW 2050, Australia; 3Liver Injury and Cancer Program, The Centenary Institute, Sydney NSW 2050, Australia

**Keywords:** liver transplant, additional operation, outcomes, graft survival

## Abstract

Advances in liver transplantation (LT) have allowed for expanded indications and increased surgical complexity. In select cases, additional surgery may be performed at time of LT rather than prior to LT due to the significant risks associated with advanced liver disease. We retrospectively studied the characteristics and outcomes of patients who underwent an additional planned abdominal or cardiac operation at time of LT between 2011–2019. An additional operation (LT+) was defined as a planned operation performed under the same anesthetic as the LT but not directly related to the LT. In total, 547 patients were included in the study, of which 20 underwent LT+ (4%). Additional operations included 10 gastrointestinal, 5 splenic, 3 cardiac, and 2 other abdominal operations. Baseline characteristics between LT and LT+ groups were similar. The median total operating time was significantly longer in LT+ compared to LT only (451 vs. 355 min, *p* = 0.002). Graft and patient survival, intraoperative blood loss, transfusion of blood products, length of hospital stay, and post-operative complications were not significantly different between groups. In carefully selected patients undergoing LT, certain additional operations performed at the same time appear to be safe with equivalent short-term outcomes and liver graft survival as those undergoing LT alone

## 1. Introduction

Liver transplantation (LT) is the treatment of choice for select patients with end-stage liver disease, hepatocellular carcinoma (HCC), and fulminant liver failure. Published short- and long-term outcomes of LT recipients are excellent with five-year and ten-year patient survival rates of 86% and 75%, respectively [1].

Enhanced survival rates of LT recipients are a reflection of the advancements in surgical techniques, anesthesia, and immunosuppressant regimens [2,3,4]. This has led to an expansion of LT indications to include recipients previously ineligible due to advanced age and comorbidity [3,5,6,7]. The complexity of LT patients has increased as more patients with non-hepatic organ dysfunction and other comorbidities are considered suitable transplant candidates [2,4]. Moreover, older patients are the fastest growing subpopulation of LT recipients and pose unique challenges due to reduced physiological reserve and greater comorbidity burden [2,3,4,5].

As LT indications expand, an increasing proportion of LT candidates have pre-existing conditions that require surgical management besides the LT itself. Patients with cirrhosis and portal hypertension undergoing abdominal or cardiac surgery have been shown to have a two- to six-fold increased mortality risk in the post-operative period in comparison to non-cirrhotic patients [8]. Compromised hepatic synthetic function places patients at greater morbidity risk due to concerns regarding hepatic decompensation, intra-operative bleeding, liver failure, ascites, and infectious complications [9,10,11]. Consequently, surgical intervention is often contraindicated in patients with advanced liver disease [9,11]. Instead, these operations may only be attempted at the time of LT or after the patient has recovered from LT. In some situations, patients require a preceding operation (e.g., cardiac valve replacement) in order to safely undergo LT.

The outcomes of patients receiving an additional operation at the same time as LT have not been extensively studied, with the majority of current literature limited to small case series focusing on specific operations only [12,13,14,15,16,17,18,19]. This study aims to compare patients who underwent an additional operation at the same time as their LT to patients who underwent LT alone in terms of donor and recipient characteristics as well as short-term and long-term outcomes.

## 2. Materials and Methods

### 2.1. Patients

Consecutive deceased donor LT recipients aged ≥18 years from 1 July 2011 to 25 July 2019 at our state-wide LT referral center were retrospectively studied. Patients were grouped into those undergoing LT only (LT only group) and those undergoing an additional planned non-hepatic operation at time of LT (LT+ group). An additional operation was defined as an abdominal or cardiac surgical procedure that was planned pre-operatively and performed under the same anesthetic as the LT but was not directly related to the LT itself. Combined liver-kidney transplant, simple hernia repair, adhesiolysis or appendicectomy, and unplanned emergent operations were excluded from the study.

### 2.2. Clinical Data

Patient demographic, clinical, operative, and laboratory data results were obtained from a prospective LT database and electronic medical records. Patient pre-transplant Model for End-Stage Liver Disease (MELD) and Donor Risk Index (DRI) scores were calculated as described previously [20,21]. The study protocol was conducted according to the Declaration of Helsinki and was approved by the Sydney Local Health District Human Ethics Research Committee with a waiver of informed consent (X19-0303).

### 2.3. Outcomes

The primary outcomes of interest were liver graft survival defined as time to death or re-transplantation and patient survival. Secondary endpoints were operative time, intraoperative blood loss, blood transfusion requirement, unplanned return to operating theater, hospital and intensive care unit (ICU) lengths of stay, hospital readmission, early allograft dysfunction (EAD) as defined by Olthoff et al. criteria [22], and acute kidney injury (AKI) defined by the Acute Kidney Injury Network (AKIN) classification [23].

### 2.4. Statistical Analysis

Continuous variables were expressed in mean ± standard deviation (SD) or median (interquartile range (IQR)) as appropriate. Differences between subgroups were analyzed using χ2 or Fisher’s exact test for categorical variables and Student’s t test, Mann–Whitney test, or one-way ANOVA for continuous variables as appropriate. Statistical analysis was performed by Statistical Package for Social Science (SPSS version 23.0, Armonk, NY, USA). A result was considered statistically significant if *p*
≤ 0.05.

## 3. Results

### 3.1. Patient Characteristics

During the study period, 573 patients received a liver transplant. Twenty patients were excluded as they received a combined liver-kidney transplant, and a further six patients received an additional emergent operation. A total of 547 patients were included in the study, of which 527 patients underwent LT only (96%) and 20 underwent LT+ (4%). The proportion of patients with LT+ fluctuated over the study period (1%–6%), although not statistically significant. Patient characteristics of LT only and LT+ groups are presented in Table 1. 

Compared to the LT only group, LT+ patients were younger (median age 46 vs. 54 years) with lower pre-transplant MELD scores (median 13 vs. 19), and lower DRIs (median 1.5 vs. 1.6); however, these differences were not statistically significant. Other clinical characteristics such as sex, proportion of re-transplanted patients, inpatient status at time of transplant and use of donation after circulatory death or split grafts were similar between patient groups. The most common additional operations were colectomy (five patients), cardiac surgery (three patients), splenectomy (three patients), and sleeve gastrectomy (two patients, Table 2). All three cardiac operations were performed prior to LT and involved cardiopulmonary bypass. The majority of the remaining additional operations were performed after LT completion and performed by the same transplant team (Table 2). The same anesthetic team was used throughout each LT+ additional operation with similar anesthetic management approaches implemented across all patients. The majority (13/20, 65%) of additional operations were performed during normal working hours (8 am–5 pm).

### 3.2. Liver Graft and Patient Survival Outcomes

During a median follow-up period of 36 months (IQR 13–62 months), there were 67 deaths and 20 re-transplants. Rates of death or re-transplantation were 78/527 (15%) in LT only and 4/20 (20%) in LT+ groups, respectively (*p* = 0.980). Graft survival at one- and three-years post-LT was 90% and 85% in the LT only group vs. 87% and 87% in the LT+ group, respectively. Patient survival at one- and three-years post-LT was 92% and 88% in the LT only group vs. 92% and 92% in the LT+ group, respectively. By Kaplan–Meier analysis, liver graft survival was similar between the groups (log rank *p* = 0.818, Figure 1A). Patient survival also did not differ between the groups (log rank *p* = 0.851, Figure 1B). Rates of EAD were similar with 129/524 (25%) and 3/20 (15%) in LT only and LT+ groups (*p* = 0.325, Table 3).

### 3.3. Operative Outcomes

Patients receiving LT+ had a longer median operative time compared to LT only patients (451 min vs. 355 min, *p* = 0.002, Table 4). There were no significant differences in the amount of intraoperative blood loss nor transfusion requirements across the groups. Similarly, there was no difference in the frequency of unexpected returns to the theater with rates of 109/527 (21%) and 5/20 (25%) in LT only and LT+ groups, respectively (*p* = 0.644). Prevalence of AKIN of any grade at 24 and 48 h was comparable between the two groups (Table 3).

### 3.4. Hospital Lengths of Stay and Readmissions

The length of the hospital admission, length of ICU admission, and number of ICU admissions prior to discharge were not significantly different between LT only and LT+ groups (Table 4). The rate of readmission within both the 30- and 90-day period was similar between LT only and LT+ patients (30% vs. 45% at 30 days, *p* = 0.154; 46% vs. 65% at 90 days, *p* = 0.091).

## 4. Discussion

The evolution of surgical techniques has allowed for LT candidates to undergo a concurrent non-hepatic planned operation at time of LT for management of other pre-existing medical conditions. With few published studies on the outcomes of such patients, it is unclear if performing an additional abdominal or cardiac operation during LT is a safe clinical decision in regard to surgical and short-term survival outcomes. Our study followed 547 adult patients undergoing deceased donor LT at a single center, of which 20 (4%) received LT+. With the exception of a longer operative time, patients undergoing LT+ had similar liver graft and patient survival outcomes to those patients undergoing LT only. 

Patients who underwent splenectomy at time of LT have been previously investigated with mixed results. Older studies, prior to the year 2000, demonstrated that these patients were at an increased risk of post-operative complications [15,24]. In contrast, a 2015 study of 40 patients undergoing LT with additional splenectomy showed equivalent rates of post-operative complications to LT only patients [25]. This is mirrored in our study which showed similar post-operative complications for LT+ and LT only patients in terms of EAD, AKI, unexpected return to theater, re-transplant, and mortality rates (*p* > 0.05 for all), this is likely a reflection of our highly selected population and advancements in surgical techniques in the last two decades. Of note, our LT+ cohort included three cardiac operations which were unique due to their surgical complexity, involvement of an artificial cardiopulmonary bypass, and timing prior to LT. Aside from hemoserous pericardial effusions in two out of three patients requiring return to theater, these patients had good short- and long-term outcomes post-transplant.

Furthermore, the median length of hospital stay was equivalent between the patient groups (18 days for both LT only and LT+ patients, *p* = 0.904). This result is supported by two previous studies of patients undergoing simultaneous cardiac surgery or splenectomy at time of LT with similar hospital stays observed ranging from 23–26 days post-operatively [16,25].

Our study showed a longer median operative time in LT+ patients in comparison to LT only (451 vs. 355 min, *p* = 0.002). Previous studies showed a slightly longer, albeit statistically insignificant, median operative time in patients undergoing an additional splenectomy or sleeve gastrectomy at time of LT by 34–48 min [12,14,25,26]. The statistically significant difference in operative time between our LT+ and LT only patients is likely a reflection of the greater range and complexity of additional operations included in the study (e.g., colectomies, coronary artery bypass graft, cardiac valve replacements, and Whipple procedure). Our LT+ patients did not have greater intraoperative blood loss nor blood product replacement in comparison to LT+ patients—a finding supported by prior studies [14,25].

Despite the greater complexity and risk associated with performing multiple simultaneous operations on patients with significant comorbidities, the LT+ patients had similar outcomes to the LT only patients. These equivalent outcomes are most likely attributed to the highly selected patient population, echoed in the smaller proportion of LT+ patients from our total cohort in comparison to the existing literature (4% vs. 5.6%–39% rates of additional operation patients in other studies) [14,15,24,25]. Factors considered in the selection of LT+ patients include their fitness as a surgical candidate, burden of comorbidities, quality of the donor liver, and the complexity of the LT operation performed immediately prior to the additional operation. This selection bias was unable to be captured by the variables measured in this study; however, it is interesting to note that the LT+ patients had numerically younger age, lower pre-transplant MELD, and lower DRI scores compared to LT only patients, albeit not reaching statistical significance.

Our findings demonstrate the potential for additional operations at time of LT to be incorporated into wider practice to manage patients with co-existing surgical problems. The benefits of combining operations allow for a single anesthetic induction, fewer surgical incisions, and one recovery process for the patient. Furthermore, concurrent operations eliminate a more hostile operative field due to subsequent adhesions and altered anatomy if the patient were to undergo separate operations [27,28]. Three patients in our cohort underwent an additional cardiac operation in order to optimize significantly impaired cardiac function which would have otherwise been a contraindication to LT. Carefully planned additional operations such as these could effectively broaden the scope of potential transplant candidates whom were previously contraindicated due to their comorbidities.

There may also be some medium- to long-term benefits of the simultaneous LT and additional operation approach in select patients. Two patients in our cohort underwent total colectomy due to active colitis at the same time as LT for primary sclerosing cholangitis (PSC). There is evidence to suggest active inflammatory bowel disease (IBD) at time of LT is an independent predictor of graft failure with a ten-fold increased risk compared to patients with inactive IBD [29]. Although data are mixed, concomitant IBD has been reported to be associated with recurrent PSC, thrombotic events (hepatic artery and portal vein thrombosis), and cellular rejection (acute and chronic) post-transplant compared to PSC patients without IBD [30]. Therefore, removal of colon with active IBD at time of LT rather than later may prevent these complications from occurring in the interim, although this approach has not been evaluated in a randomized controlled trial. An additional two of our patients underwent simultaneous LT and sleeve gastrectomy. Zamora-Valdes et al. recently demonstrated that simultaneous LT and sleeve gastrectomy resulted in more effective sustained weight loss three years post-transplant compared to LT alone [27]. The greatest difference was seen during the first 6–12 months when patients who received combined LT and sleeve gastrectomy continued to lose weight while those with LT alone regained weight to almost baseline values. Hence, the combined approach may abrogate this early weight gain (compounded by induction corticosteroids and dietary changes) and its metabolic complications. If not, the authors also argue that plans for subsequent sleeve gastrectomy post-transplant may be thwarted by complications such as rejection, infection, or disease recurrence [27].

The potential benefits of performing additional surgery at the time of LT needs to be weighed against the disadvantage of performing multiple operations where there are increased demands on scheduling, resources, and personnel (e.g., multiple surgeons) to be available often at short notice for the arrival of the donor liver. It is important to note, although our study demonstrates the feasibility and safety of performing an additional operation at the same time as LT, it does not address the question of whether this approach is better than delaying the additional operation until after the LT. In our study, the additional operation of three patients (all cardiac) was needed in order to safely proceed with LT and could not be delayed.

There are several limitations to our study. Firstly, the retrospective nature of the study inherently relies on the level of accuracy and completeness of the data found in medical records. However, all collected data points were based on hard endpoints (e.g., death, re-transplant, blood loss, blood products used, and length of stay) to minimize bias and subjectivity in data collection. As aforementioned, there is selection bias in choosing patients suitable for an additional operation beforehand. We do not know how many (if any) patients were refused LT because of the need for complex additional surgery. This bias can only be eliminated by performing a randomized control trial which is neither medically nor ethically feasible. Finally, this is a single center study with a relatively small numbers of patients receiving a heterogenous mix of additional operations. Therefore, it is difficult to extrapolate our findings to other centers and larger multi-center studies are required for confirmation.

In conclusion, our single-center Australian experience shows that certain planned abdominal and cardiac operations can be performed safely at the same time as LT in carefully selected patients with equivalent short-term outcomes and liver graft survival observed compared to those who undergo LT alone.

## Figures and Tables

**Figure 1 jcm-09-00608-f001:**
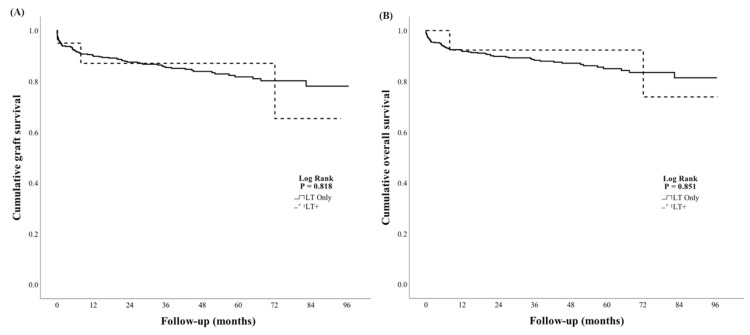
Survival analyses. Kaplan–Meier analyses of cumulative liver graft survival (**A**) and overall survival (**B**) in liver transplantation only (LT only) and liver transplantation with an additional operation (LT+) groups.

**Table 1 jcm-09-00608-t001:** Demographic and clinical characteristics.

	LT Only *n* = 527	LT+ *n* = 20	*p*
Number of transplants per year group (%)			0.062
2011–2013	139 (96)	6 (4)	
2014–2016	209 (99)	3 (1)	
2017–2019	179 (94)	11 (6)	
Male (%)	375 (71)	16 (75)	0.390
Median age (IQR)	54 (47–59)	46 (38–60)	0.131
Primary indication for LT (%)			0.601
HCC	142 (27)	5 (25)	0.847
HCV	102 (19)	4 (20)	0.943
Alcoholic liver disease	71 (13)	5 (25)	0.143
NAFLD	40 (8)	2 (10)	0.691
ALF	3 (1)	0 (0)	0.739
Others	169 (32)	4 (20)	0.255
Re-transplant patient (%)	27 (5)	2 (10)	0.339
Inpatient at time of transplant offer (%)	184 (35)	6 (30)	0.650
Median pre-transplant MELD score (IQR)	19 (14–25)	13(10–25)	0.141
DCD donor (%)	39 (7)	1 (5)	0.686
DRI (%)	1.6 (1.4–1.9)	1.5 (1.4–1.7)	0.192
Split graft (%)	63 (12)	3 (15)	0.681

The data are shown in number (percentage) and median (interquartile range). ALF, acute liver failure; DCD, donation after circulatory death; DRI, donor risk index; HCC, hepatocellular carcinoma; HCV, hepatitis C virus, IQR, interquartile range; MELD, model for end-stage liver disease; NAFLD, non-alcoholic fatty liver disease; LT+, liver transplantation with an additional operation; LT only, liver transplantation only.

**Table 2 jcm-09-00608-t002:** Additional operation characteristics.

Additional Operation	*n* = 20 (%)	Timing in Relation to Implantation of the Donor Liver	Surgical Team
Abdominal operation	17 (85)		
*Gastrointestinal*			
Total colectomy for Crohn’s colitis	2 (10)	After	Transplant
Hemicolectomy for villous adenoma with dysplasia	2 (10)	After	Colorectal
Sleeve gastrectomy for obesity	2 (10)	After	Transplant
Small bowel resection of jejunal stromal tumor	1 (5)	After	Transplant
Subtotal colectomy and ileostomy for colonic adenomas ^#^	1 (5)	After	Colorectal
Resection of ampullary hamartoma	1 (5)	Before	Transplant
Ileocolic resection and formation of stoma for Crohn’s stricture of terminal ileum	1 (5)	After	Transplant
*Splenic **			
Splenectomy for splenic artery aneurysm at hilum	2 (10)	1 Before, 1 After	Transplant
Splenic artery aneurysm ligation/excision	2 (10)	Before	Transplant
Splenectomy for prevention of antibody mediated rejection in a re-transplanted patient	1 (5)	Before	Transplant
*Other*			
Whipple procedure for hilar cholangiocarcinoma	1 (5)	After	Transplant
Partial nephrectomy for renal cell carcinoma	1 (5)	Before	Urology
Cardiac operation	3 (15)		
Cardiac valve replacement for severe tricuspid regurgitation	1 (5)	Before	Cardiothoracic
Cardiac valve replacement for severe aortic regurgitation	1 (5)	Before	Cardiothoracic
Coronary artery bypass graft and atrial septal defect closure	1 (5)	Before	Cardiothoracic

^#^ Patient also had combined liver and kidney transplant. * All splenectomy patients received appropriate vaccinations pre-transplant and prophylactic antibiotics post-transplant.

**Table 3 jcm-09-00608-t003:** Early allograft dysfunction and acute kidney injury outcomes.

	LT Only	LT+	*p*
EAD * (%)	129 (25)	3 (15)	0.325
AKIN 1 at 24 h (%)	122 (23)	6 (30)	0.487
AKIN 2 at 24 h (%)	77 (15)	2 (10)	0.557
AKIN 3 at 24 h (%)	38 (7)	1 (5)	0.700
AKI any at 24 h (%)	237 (45)	9 (45)	0.984
AKIN 1 at 48 h (%)	105 (20)	3 (15)	0.572
AKIN 2 at 48 h (%)	91 (18)	5 (25)	0.389
AKIN 3 at 48 h (%)	65 (13)	2 (10)	0.739
AKI any at 48 h (%)	262 (50)	10 (50)	0.980

The data are shown in number (percentage). AKI, acute kidney injury; AKIN, acute kidney injury network criteria; EAD, early allograft disfunction; h, hours; LT+, liver transplantation with an additional operation; LT only, liver transplantation only. * Olthoff et al. [22] criteria.

**Table 4 jcm-09-00608-t004:** Surgical outcomes.

	LT Only	LT+	*p*
Total operative time in minutes (IQR)	355 (289–425)	451 (410–518)	0.002
Estimated blood loss in mL (IQR)	3678 (2000–6183)	4300 (2000–9000)	0.625
Units of blood products transfused intraoperatively (IQR)			
PRBC	4 (1–8)	5 (0–12)	0.748
FFP	6 (2–10)	4 (2–13)	0.691
Platelets	1 (0–2)	0 (0–2)	0.416
Cryoprecipitate units	8 (0–15)	8 (0–10)	0.532
Cellsaver (mL)	633 (220–1290)	821 (230–1544)	0.756
Unplanned return to theater (%)	109 (21)	5 (25)	0.644
Hospital stay post-transplant (IQR)	18 (13–27)	18 (14–29)	0.904
Total ICU days (IQR)	6 (4–10)	6 (5–10)	0.919
Number of ICU stays (IQR)	1 (1–1)	1 (1–1)	0.657
Readmission in 30 days (%)	151 (30)	9 (45)	0.154
Readmission in 90 days (%)	227 (46)	13 (65)	0.091

The data are shown in number (percentage) and median (interquartile range). FFP, fresh frozen plasma; ICU, intensive care unit; LT+, liver transplantation with an additional operation; LT only, liver transplantation only; PRBC, packed red blood cells.

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
