# Peer review of "Impact of Having a Planned Additional Operation at Time of Liver Transplant on Graft and Patient Outcomes"

_jcm, 2020, doi:10.3390/jcm9020608_

Round 1
Reviewer 1 Report
This result was so interesting, however I have some comments regarding this content.
Please add the information regarding surgical details. Cardia surgery was conducted after liver transplantation? Liver transplantation was first in all cases in method? Other additional surgery was conducted by same surgical team? Please tell their anesthesia information. They used same management regarding intraoperative anesthesia management? I felt that cardiac surgery was quite different among other additional surgery? They used artificial cardiopulmonary device, which might cause much bleeding? What factors influenced their excellent outcome compared to other manuscript? Please discuss in discussion.Author Response
Please see the attachment

Reviewer 2 Report
I like this paper, which shows the feasibility of the additional surgical procedures performed at time of LTx in liver transplant recipients. I would congratulate the authors very good results in LT+ group, when compare to LT only group. The indications for additional operations are very heterogenous. It is difficult to compare the extensive cardiac surgery to small bowel resection (for instance) as additional procedures in terms of the risk of morbidity and mortality. The authors correctly point out the limitations of the study and the bias and subjectivity in data collection.
